# MARS - A Foundational Map Auto-Regressor

**Qi Zhang\*, Suvam Bag\*, Rupanjali Kukal\*,**
**Mikael Figueroa, Rishi Madhok, Nikolaos Karianakis, Fuxun Yu**
Microsoft
qizhang5@microsoft.com

## ABSTRACT

Map generation tasks feature extensive non-structural *vectorized data* (e.g., points, polylines, and polygons) and thus pose significant challenges to common pixel-wise generative models. Conventional approaches use multiple stages, first segmenting these features at the pixel level and then performing vectorized post-processing, with errors and complexity compounding at each stage. Motivated by the recent success of auto-regressive language modeling, we propose the first map foundation model, named Map Auto-Regressor (MARS), that is capable of generating both multi-polyline road networks and polygon buildings in a unified manner. For training MARS, we collected to our knowledge the largest multi-class map extraction dataset totaling 3.4M examples, which we call MAP-3M. Across four road and building datasets, MARS outperforms or matches the performance of multistage baselines. Additionally, we develop a "Chat with MARS" capability that enables interactive human-in-the-loop map generation and correction, supported by the auto-regressive nature of our end-to-end approach. We release our MAP-3M dataset and project demo page at (1) https://huggingface.co/datasets/bag-lab/MAP-3M and (2) https://huggingface.co/spaces/bag-lab/MARS, respectively.

.

## 1 INTRODUCTION

Automatically generating maps from overhead or remote-sensing imagery involves converting *rasterized* pixels to *vectorized* geometric primitives including points, polylines, and polygons, which represent diverse map elements such as roads, buildings, and water bodies. A key challenge for map generative modeling lies in the *vectorized* representation of maps (Congalton, 1997; Jiang et al., 2024). Most current visual generative tasks are *rasterized*, i.e., producing pixel grids, leveraging off-the-shelf tensor-based generative architectures such as the Segment-Anything Model (SAM; Kirillov et al., 2023) and diffusion models (Rombach et al., 2022). In contrast, map elements are geometric, vector shapes, often consisting of a variable, unstructured set of points, polylines, and polygons, as shown in Fig. 1 (c). This structural mismatch poses unique difficulties for generic map generative architectures.

As a workaround, many papers tackle map generation by first applying rasterized segmentation and then vectorized post-processing in a multistage pipeline (Xu et al., 2023). For example, SAM plus post-processing methods have been explored for road network graph extraction (Kirillov et al., 2023; Hetang et al., 2024). Similarly, Wang et al. (2024) use SAM and design different post-processing logic for building delineation.

Such methods suffer from two major drawbacks that prevent them from serving as a foundation model for map generation: (1) limited generality across map classes and (2) suboptimal performance. For the former, post-processing (e.g., key-point and edge extraction) is typically heuristic-based, and the heuristics differ substantially between feature classes, e.g., road networks (multi-polylines with complex intersections) versus buildings (non-overlapping polygons). As a result, these methods generally support only a single map feature type. For the latter, because the generation is not learned end to end within a unified architecture, performance is often constrained. It

also leads to more hyperparameters, such as NMS IoU thresholds for key-point de-duplication or confidence thresholds for edge connections.

In this work, we address map generation tasks by treating *vectorized map primitives as a formal language*. Our approach is inspired by sequence-to-sequence learning (Radford et al., 2018), where auto-regressive architectures have demonstrated effectiveness in producing complex non-structural outputs. Specifically, we design a novel unified *map-to-sequence* framework to convert all vectorized map primitives (roads and buildings) into a sequential language representation. Then, we propose an end-to-end *map auto-regressor (MARS)* architecture to conduct sequence-to-sequence map primitive learning. To ensure generalization and scalability, we do not deviate from conventional standards for the MARS architecture and use only a transformer-based vision encoder and an auto-regressive decoder, without any intermediate or post-processing being applied.

Without bells and whistles, MARS learns to generate all types of map elements at scale. To the best of our knowledge, *this is the first map foundation model* capable of generating both vectorized roads and buildings within a single end-to-end model without any post-processing, marking an important step toward scalable and generalizable map generation.

Benefiting from the auto-regressive nature of our model, we further identify an emergent *prompt-following capability* of MARS, like that of GPT (Radford et al., 2018). We then develop a "**Chat with MARS**" feature, where human users can prompt MARS by giving a starting point of a missing street or a building, which MARS will then auto-complete. This auto-completion of target objects enables a new interactive map generation capability: *human-in-the-loop map generation and correction*.

Training scalable foundation models relies on large-scale data, but all current map datasets are both single class and limited in quantity. We thus curate **MAP-3M**, the largest dataset for map generation to date. This collection contains three million images with high-quality annotations for both roads and buildings, **10×** larger in the number of images and **100×** broader in geographic coverage than common benchmarks such as Cityscale (He et al., 2020a), SpaceNet3 (SpaceNet, 2018) and AiCrowd (AICrowd, 2020).

Overall, our main contributions are as follows:

- We proposed a foundational map auto-regressive model architecture: MARS, that can end-to-end generate various vectorized map elements without any post-processing.

- We identified and developed "Chat with MARS" that enables brand-new interactive human-in-the-loop map generation and correction capability.

- We curated by-far the largest multi-class map generation dataset to facilitate the foundational MARS training. Dataset has been released.

We conduct extensive benchmarks on MAP-3M and various downstream applications. Experimental results demonstrate the superior performance of our unified architecture compared to previous state-of-the-art models, laying the foundation for future advancements in map generation.

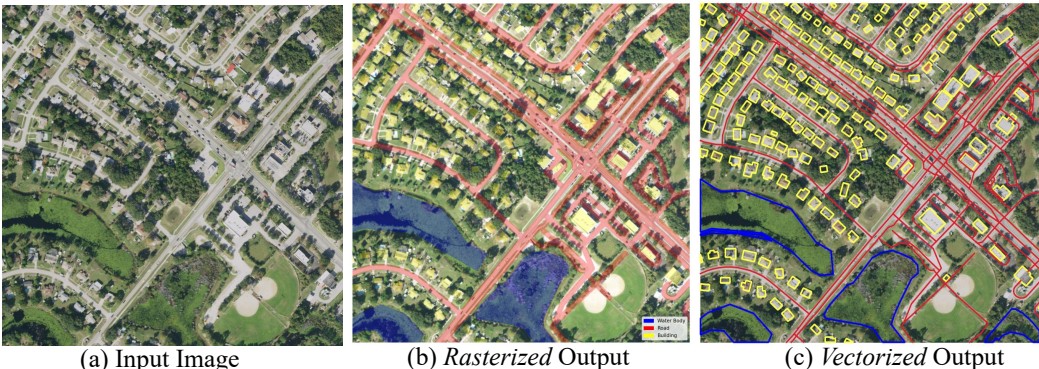

(a) Input Image  (b) *Rasterized* Output  (c) *Vectorized* Output

Figure 1: Examples of rasterized and vectorized map generation. Source: MARS.

## 2 THE MARS FRAMEWORK

Our MARS framework consists of two main parts: a *map-to-sequence* algorithm that transforms vector data into a sequential format, and an end-to-end *map auto-regressor* architecture, which includes a vision backbone and an auto-regressive transformer connected with cross-attention.

### 2.1 MAP-TO-SEQUENCE CONVERSION

To start with, all map objects (building, roads, water bodies, etc.) can be categorized into three basic types: **Point**, **Polyline**, and **Polygon**. For each object, they can be represented by a series of vertices and/or edges. For example, a point can be represented as a tuple of its coordinates: $[x, y]$. A polyline can be then represented as a sequence of points: $[x_1, y_1, x_2, y_2, x_3, y_3, x_4, y_4, ...]$, where $x_i, y_i$ is the $i$-the vertex's coordinates. A polygon is a closed-loop polyline, in which the sequence ends with the starting vertex, e.g., $[x_1, y_1, x_2, y_2, x_3, y_3, x_4, y_4, ..., x_1, y_1]$.

With above sequential representation for single object, representing multiple *Polygons* without intersections such as multiple buildings can be easily formulated as below:

$$[\, B, x_1^1, y_1^1, x_2^1, y_2^1, x_3^1, y_3^1, ..., x_1^1, y_1^1, \quad B, ..., \quad B, x_1^i, y_1^i, x_2^i, y_2^i, x_3^i, y_3^i, ..., x_1^i, y_1^i, \quad ... \\ B, x_1^N, y_1^N, x_2^N, y_2^N, x_3^N, y_3^N, ..., x_1^N, y_1^N\,], \tag{1}$$

where $B$ is a class token for building objects, $i$ denotes the $i$-th building, and $x_j^i, y_j^i$ is the $j$-th vertex's coordinates in the $i$-th building, and $N$ is the number of buildings.

Representing intersectional *Multi-Polyline* road networks as a sequence poses the greatest challenge, as they form highly complex graphs, with intersections, merges and roundabouts, as shown in Fig. 2 (b). This poses many challenges for standard graph traversal methods (Christofides, 1973) to account for the semantic continuity and/or separability of those roads.

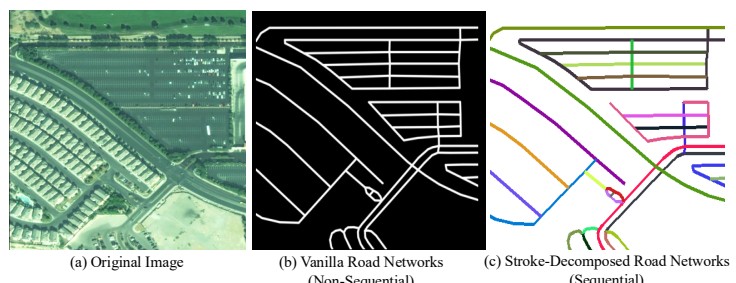

(a) Original Image    (b) Vanilla Road Networks (Non-Sequential)    (c) Stroke-Decomposed Road Networks (Sequential)

Figure 2: Our stroke-based algorithm can decompose a non-sequential flat road networks to multiple single polylines, which can then be sequentialized for auto-regressive learning. Source: MARS.

To sequentialize such road graphs, we adopt a stroke-based algorithm (Yan et al., 2024). This algorithm essentially decomposes all road segments by intersection points (which have an edge degree >3), and then merge consecutive segments within certain angle tolerance (e.g., $< 30°$) to be a single road. This simplifies complex multi-polyline networks into multiple single-polyline roads that align with real-world road definitions, as shown by different colored single polylines in Fig. 2 (c). The multiple single-polylines can be then represented using the same formulation as Eq. 1, with the specialized token replaced by $R$ to denote roads.

With this map-to-sequence framework, we thus enable converting all map elements as sequences (points, polygons, multi-polylines). With multiple object types on the map, we just need to construct different class tokens, and then combine them into the final sequence:

$$[\, P, x_1^0, y_1^0, \quad ..., \quad B, x_1^1, y_1^1, x_2^1, y_2^1, x_3^1, y_3^1, ..., x_1^1, y_1^1, \quad ..., \\ R, x_1^i, y_1^i, x_2^i, y_2^i, x_3^i, y_3^i, ..., \quad W, x_1^N, y_1^N, x_2^N, y_2^N, x_3^N, y_3^N, ..., x_1^N, y_1^N \,], \tag{2}$$

where $P, B, R, W$ are the specialized token for Points, Buildings, Roads, Waterbodies, etc., and $i$ denotes the $i$-th object, and $x_j^i, y_j^i$ is the $j$-th vertex's coordinates in the $i$-th object. By our map-to-sequence framework, we can convert a vectorized map tile as a sequence, which enables auto-regressive end-to-end map learning without any manual designed vectorized post-processing.

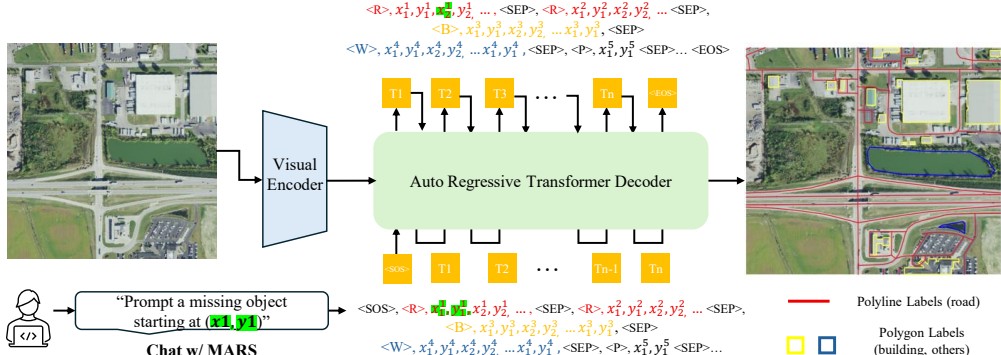

Figure 3: MARS unifies vectorized map generation by end-to-end visual auto-regressive modeling. We adopt Swin Transformer as vision encoder to extract visual context features. The auto-regressive transformer cross-attends visual context features with map tokens, and then generates the sequences (points, polylines, polygons) in an auto-regressive manner. Benefited from teacher-forced training, we can also support human-in-the-loop map generation with prompts. Image source: MARS.

## 2.2 MAP AUTO-REGRESSOR ARCHITECTURE

Figure 3 shows the MARS architecture composed of two major parts: (1) a vision backbone to extract the visual features, and (2) an auto-regressive transformer for vectorized map generation.

The vision backbone (e.g., Swin-L (Liu et al., 2021)) is used to process the satellite imagery and fuse visual context feature to the autoregressive decoder. To pass the multilevel features from the encoder to the decoder, we design a minimal convolutional feature aggregator to concatenate features from four hierarchical levels of the backbone with adaptive upsampling. A feature bridge flattens the visual features and adopts cross-attention to fuse the image context feature into the decoder.

Built on a vanilla transformer (Radford et al., 2018), the decoder generates the semantic class tokens and coordinate tokens of each map object auto-regressively by using the encoder's visual context features together with previously generated tokens. It uses single-directional causal attention, ensuring that each token attends only to past tokens. In the training phase, the autoregressive decoder is trained by teacher forcing (Radford et al., 2018). For optimization, we apply a standard cross-entropy loss to supervise the learning of both semantic class tokens and coordinate tokens. Additionally, we sort all map objects by their distance to the image centroid, and when distances are similar, we sort them by their clockwise angle. This creates a consistent spiral ordering across the dataset and ensures reproducible decoding. By jointly supervising labels and discretized coordinates under the same loss function, the model learns to generate complete map object sequences in a consistent and unified manner. In inference phase, the decoder autoregressively predicts tokens from a start-of-sequence token until an end-of-sequence token.

To represent both semantic categories and spatial positions in a unified manner, we construct a shared decoding vocabulary $D \in \mathbb{R}^{B_o + B_c}$, where $B_o$ denotes the number of semantic classes in the ontology and $B_c$ denotes the number of pixel locations along the image height/width dimensions (e.g., 224). For simplicity and consistency, we share the coordinate tokens for the width ($x$) and height ($y$) dimensions. The shared decoding vocabulary $D$ is further extended to include special tokens, such as the start-of-sequence token, separator token, pad token, and end-of-sequence token.

## 3 *Chat with MARS*: HUMAN-IN-THE-LOOP MAP GENERATION

With the autoregressive teacher-forcing training, the prediction of each token is conditioned on all preceding tokens. *Prompt following capability* thus emerges in MARS, which naturally allows user interventions to be seamlessly integrated into the decoding process. We thus develop *"Chat with MARS"* capability, an interactive human-in-the-loop paradigm for collaborative map generation.

Chat-with-MARS integrates user prompts through three complementary modes of interaction: start-of-sequence chatting, mid-sequence chatting, and end-of-sequence chatting. These modes enable

diverse forms of intervention during decoding, allowing users to indicate a missing object, correct drifting errors, and enhance performance in complex geospatial scenes.

### 3.1 START OF SEQUENCE CHATTING

Start-of-sequence (SOS) chatting aims to provide the first map element's starting point so that MARS generates maps with a better conditioned starting point. As shown in the highlighted part in Eq. 3, one class token and two coordinate tokens $(x, y)$ from one user click could be passed in as prompt to better guide the following auto-regressive token generation:

$$[\langle SOS \rangle, B, x_1^1, y_1^1, x_2^1, y_2^1, x_3^1, y_3^1, ..., x_1^1, y_1^1, ... \langle EOS \rangle]. \tag{3}$$

This is particularly helpful when a test image is extremely blurry or out-of-domain: once the first vertex predicted by MARS is ill-conditioned, due to error accumulation of auto-regressive nature, the whole sequence may suffer from less detections. In such case, SOS chatting can greatly improve the full image prediction performance. Visualizations could be found in Fig. 4 (a-d).

### 3.2 MID OF SEQUENCE CHATTING

Mid-of-sequence (MOS) chatting aims to intercept MARS's prediction sequence when it drifts from the desired trajectory, which is common in vectorized road generation. As shown in the highlighted part in Eq. 4, two coordinate tokens $(x, y)$ from one user click could be prompted to replace the old drifting tokens and redirect the following road generation:

$$[\langle SOS \rangle, R, x_1^1, y_1^1, x_2^1, y_2^1, \cancel{x_{old}, y_{old}}, x_3^1, y_3^1, ..., ... \langle EOS \rangle]. \tag{4}$$

This is particularly helpful when certain predictions in an image needs to be adjusted. Visualizations could be found in Fig. 4 (e-h). For minimal impact, we enforce the newly generated tokens to be within that object (i.e., stop when hitting next object class token) without affecting other objects.

### 3.3 END OF SEQUENCE CHATTING

End-of-sequence (EOS) chatting aims to augment MARS's prediction when there are objects missed from the final predictions, which is common for various small map elements. As shown in the highlighted part in Eq. 5, we can remove the $\langle EOS \rangle$ token, and then prompt with new object class token and coordinate tokens $(x, y)$ so as to resume the map generation:

$$[\langle SOS \rangle, R, x_1^1, y_1^1, x_2^1, y_2^1, x_3^1, y_3^1, ..., \cancel{\langle EOS \rangle}, B, x_1^1, y_1^1, ..., \langle EOS \rangle]. \tag{5}$$

Such EOS chatting could be used to improve recall. Visualizations could be found in Fig. 4 (i-l).

*Chat-with-MARS* can be extended with mixed forms, easily forming multi-round conversations for interactive editing. We believe it has strong potential for a wide range of real-world localized map editing and maintenance tasks such as OSM Change Analyzer (OpenStreetMap, n.d.).

## 4 MAP-3M: A LARGE-SCALE MAP DATASET

For map generation, most widely-used benchmarks in the literature cover only single-class annotations with limited image quantities, significantly restricting their utility for real-world tasks. For example, Cityscale (He et al., 2020a) and SpaceNet3 (SpaceNet, 2018) exclusively contain road annotations, while AI-Crowd (AICrowd, 2020) focuses solely on building annotations.

To overcome these constraints, we curated MAP-3M, the largest high-resolution aerial image + map dataset to date, comprising approximately 3M high-resolution images ($10\times$ bigger than the other available datasets) enriched with high-quality annotations for both buildings and roads. An comparison between MAP-3M and other literature datasets is shown in Table 1.

**Images.** MAP-3M images are sourced from the National Agriculture Imagery Program (NAIP; U.S. Department of Agriculture, 2025) and were accessed via Microsoft Planetary Computer. Leveraging population data (United States Cities Database, 2025), we evenly sample 5,000 cities from 50 states. More details are in Appendix. A.1.

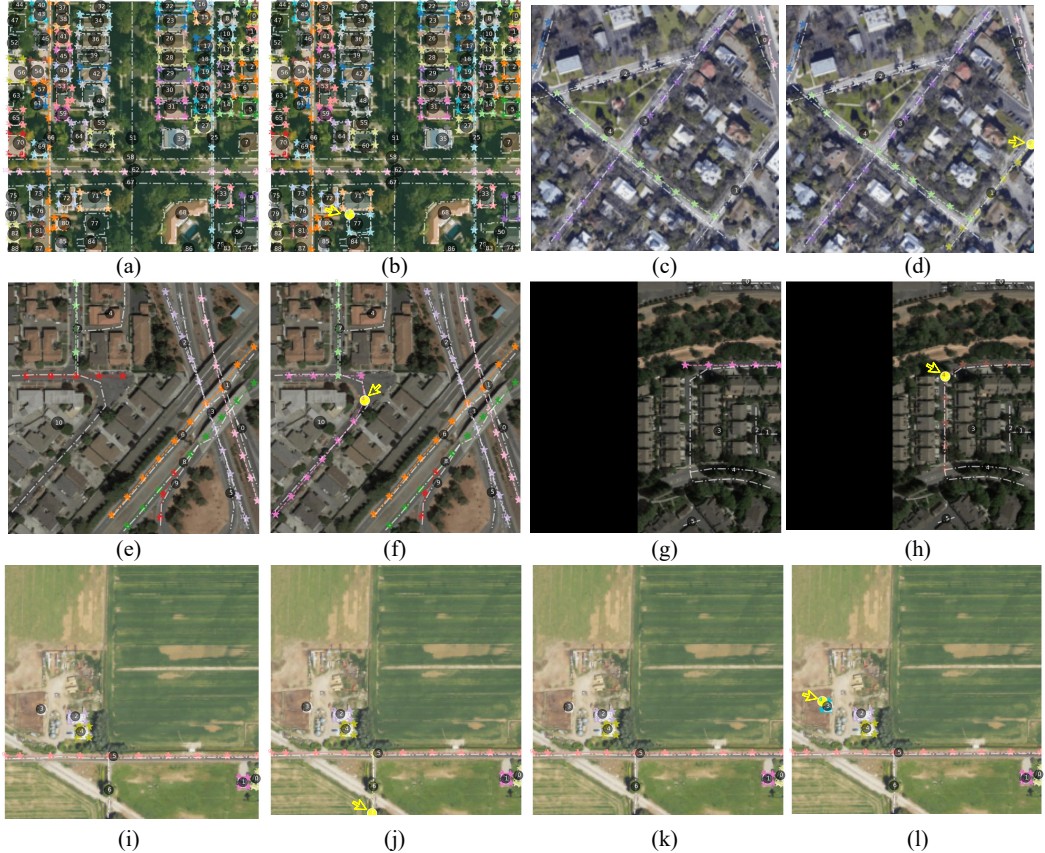

Figure 4: **Chat with MARS**. The user interactions are provided as single clicks, shown as yellow arrows. White dash-dotted line: Ground truth. Colored lines with asterisks: Predicted sequences. Black circles with white numbers: Object IDs. (a–b): Start-of-sequence interaction enables recovery of a missing building object (ID 77) in a NAIP image. (c–d): Similarly, a missing road object (ID 1) is recovered in a Cityscale image. (e–f): Mid-sequence interaction helps correct the prediction of a drifting road object (ID 10) in Cityscale. (g–h): Another mid-sequence example shows recovery of a drifting road segment (ID 3) in Cityscale. (i–j): End-of-sequence interaction allows the model to recover a missing road object (ID 6) in a NAIP image. (k–l): End-of-sequence interaction recovers a missing building object (ID 3) in the same NAIP image. Source: MARS.

Table 1: MAP-3M provides $10\times$ more images, and $100\times$ more spatial coverages than literature datasets. GSD: meter/pixel. ∗: Cityscale contains 2K chips, we tile to 224x224 for comparison.

| Dataset | # Images | Image Size | Coverage Area | GSD | Building Cls | Road Cls |
|---|---|---|---|---|---|---|
| Cityscale (He et al., 2020a) | 49220 | 224x224* | 2470 $km^2$ | 1.0 | × | ✓ |
| SpaceNet3 (SpaceNet, 2018) | 2541 | 400x400 | 407 $km^2$ | 1.0 | × | ✓ |
| AICrowd (AICrowd, 2020) | 258044 | 300x300 | 2090 $km^2$ | 0.3 | ✓ | × |
| **MAP-3M** (Ours) | ∼**3M** | 512x512 | **294069** $km^2$ | 0.6 | ✓ | ✓ |

**Labels.** MAP-3M collects vectorized annotations that cover two fundamental map classes: *buildings* and *roads*. Other classes such as waterbodies, parking lot, etc. can be sequentialized using the same representation methods (polygons/polylines) in Eq. 2 but are much less dominant, so we omit these map classes for now. Fig. 5 shows the diversity and quality of the MAP-3M annotations.

In addition to the quality, MAP-3M has the largest quantity compared to literature map datasets, as shown in Table 1. The 3M positive samples are $10\times$ larger in number of images, and $100\times$

larger in spatial area coverage than the previous largest dataset, which enables us to conduct robust foundational model pre-training and improve downstream generalization. Dataset has been released at `https://huggingface.co/datasets/bag-lab/MAP-3M`.

## 5 EXPERIMENTAL RESULTS

**Experimental setup.** For all model evaluations, we pretrained MARS on our MAP-3M dataset. For different downstream tasks, we fine-tuned pretrained MARS on these datasets and benchmarked its performance against specialized models. Three literature single-class benchmarks are used, together with our MAP-3M VAL set:

- **Cityscale** (He et al., 2020a): An urban-scale benchmark for road graph extraction, constructed from high-resolution aerial imagery.
- **SpaceNet** (SpaceNet, 2018): A benchmark for road extraction from satellite imagery, with a focus on generalization across geographies.
- **AICrowd** (AICrowd, 2020): A diverse dataset used in competitive challenges that covers building detection in complex urban scenes.
- **MAP3M-VAL**: Our MAP-3M validation set containing two classes: building and roads. MAP3M-VAL includes 5000 images separated from the train set.

### 5.1 FROM SINGLE TO MULTI-CLASS: TOWARDS UNIFIED MAP GENERATION

Figure 6 demonstrates the performance visualizations of MARS on handling multi-class geometry predictions. From simple single-polyline road, to complex multi-polylines, and to dense mixed buildings/roads, MARS learns to generate various map tiles in a unified end-to-end way.

To better understand the trade-offs and design choices in our modeling approach, we perform an ablation study comparing single-class and multi-class architectures in Table 2. As shown in Table 2, the model scales well with increasing class diversity without obvious degradation in performance. As handling multiple map feature types within a unified framework is often more practical and efficient than deploying separate models for each class, such ontology scalability of MARS shows its great potential to serve as a map foundational model for future development.

### 5.2 DOWNSTREAM FINETUNING COMPARISON

**Roads.** We employed TOPO, a topological accuracy metric (He et al., 2020b), to assess how closely the predicted road graphs align with the ground truth in terms of structural connectivity.

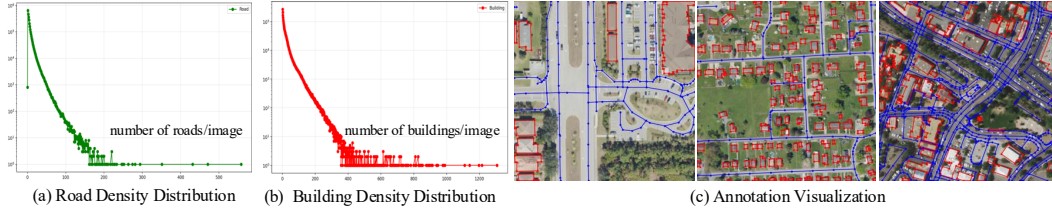

(a) Road Density Distribution        (b) Building Density Distribution        (c) Annotation Visualization

Figure 5: MAP-3M features (a-b) wide map element density distribution per image, and (c) high-quality annotations with diverse vectorized map geometry coverage. Source: MARS.

Table 2: MARS architecture unifies single- or multi-classes learning by simply adding class tokens.

| Classes | MAP3M-VAL | | | | CITYSCALE | | | SPACENET | | | AICROWD |
|---|---|---|---|---|---|---|---|---|---|---|---|
| | P | R | F1 | IOU | P | R | F1 | P | R | F1 | IOU |
| Road | 89.8 | **85.6** | **87.7** | - | **85.1** | 78.2 | 81.5 | 79.3 | 83.3 | 81.2 | - |
| Building | - | - | - | **64.4** | - | - | - | - | - | - | 95.0 |
| Both | **90.1** | 77.1 | 83.1 | 61.0 | 84.3 | **81.5** | **82.9** | **79.7** | **84.6** | **82.1** | **97.3** |

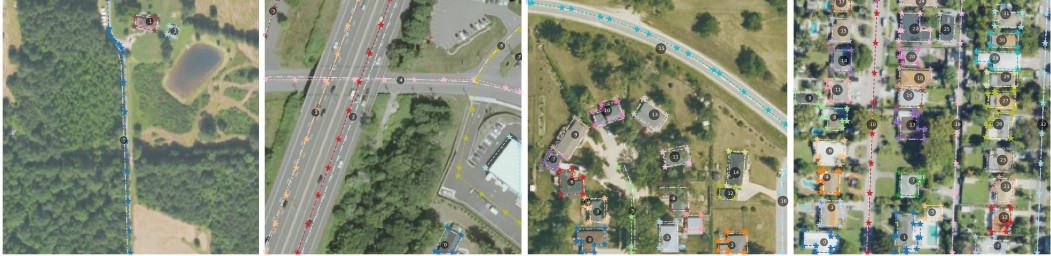

Figure 6: MARS visualization. MARS is capable of handling diverse scenes and geometries without any post-processing, demonstrating the generalization and scalability of an end-to-end learning framework. White dash-dotted line: Ground truth. Colored lines with asterisks: Predicted sequences. Black circles with white numbers: Object IDs. Source: MARS.

Table 3: TOPO-based road performance comparison on Cityscale and SpaceNet.

| Model | CITYSCALE | | | SPACENET | | |
|---|---|---|---|---|---|---|
| | P | R | F1 | P | R | F1 |
| Seg-UNet (Ronneberger et al., 2015) | 75.34 | 65.99 | 70.36 | 68.96 | 66.32 | 67.61 |
| Seg-DRM (Máttyus et al., 2017) | 76.54 | 71.25 | 73.80 | 82.79 | 72.56 | 77.34 |
| Seg-Improved (Batra et al., 2019) | 75.83 | 68.90 | 72.20 | 81.56 | 71.38 | 76.13 |
| Seg-DLA (Yu et al., 2018) | 75.59 | 72.26 | 73.89 | 78.99 | 69.80 | 74.11 |
| RoadTracer (Bastani et al., 2018) | 78.00 | 57.44 | 66.16 | 78.61 | 62.45 | 69.90 |
| Sat2Graph (He et al., 2020b) | 80.70 | 72.28 | 76.26 | 85.93 | 76.55 | 80.97 |
| TD-Road (He et al., 2022) | 81.94 | 71.63 | 76.43 | 84.81 | 77.80 | 81.15 |
| RNGDet (Xu et al., 2022) | 85.97 | 69.78 | 76.87 | 90.91 | 73.25 | 81.13 |
| RNGDet++ (Xu et al., 2023) | 85.65 | 72.58 | 78.44 | 91.34 | 75.24 | **82.51** |
| SamRoad (Hetang et al., 2024) | **90.47** | 67.69 | 77.23 | **93.03** | 70.97 | 80.52 |
| **MARS** | 84.28 | **81.53** | **82.88** | 79.68 | **84.56** | 82.05 |

Table 4: Building performance comparison on Aicrowd V1.

| Model | AICROWD-V1 | | | | | | | | | | |
|---|---|---|---|---|---|---|---|---|---|---|---|
| | AP | AP50 | AP75 | AR | AR50 | AR75 | bAP | IoU | C-IoU | PoLiS | N-ratio |
| PolyMapper | 55.7 | 86.0 | 65.1 | 62.1 | 88.6 | 71.4 | 22.6 | 77.6 | 65.3 | 2.215 | 1.29 |
| FFL* | 67.0 | 92.1 | 75.6 | 73.2 | 93.5 | 81.1 | 34.4 | 84.3 | 73.8 | 1.945 | 1.13 |
| PolyWorld | 63.3 | 88.6 | 70.5 | 75.4 | 93.5 | 83.1 | 50.0 | 91.2 | 88.2 | 0.962 | 0.93 |
| PolyBuilding | 78.7 | 96.3 | 89.2 | 84.2 | 97.3 | 92.9 | - | 94.0 | 88.6 | - | 0.99 |
| HiSup | 79.4 | 92.7 | 85.3 | 81.5 | 93.1 | 86.7 | 66.5 | 94.3 | 89.6 | 0.726 | - |
| Pix2Poly | 79.6 | 91.6 | 85.2 | 87.7 | - | - | - | 95.03 | 89.85 | **0.479** | 1.111 |
| GeoFormer | **91.5** | **96.6** | **93.1** | 97.8 | 98.8 | 98.1 | **97.1** | **98.1** | **97.4** | 0.913 | 1.01 |
| **MARS** | 87.30 | 95.20 | 90.46 | **97.94** | **99.28** | **98.62** | 92.44 | 97.32 | 96.31 | 0.997 | **0.4542** |

As shown in Table 3, our model demonstrates superior performance compared to prior state-of-the-art approaches. On the Cityscale dataset, MARS achieves the highest Recall and F1 scores, outperforming RNGDet++ (Xu et al., 2023) by a notable margin, from **78.44%** to **82.88%** in terms of F1. Similarly, on SpaceNet, MARS achieves a leading recall of **84.56%**, and only **-0.46%** in F1 compared to the best priori method, RNGDet++ (Xu et al., 2023).

Of note, all prior methods in Table 3 are road-specialized models rather than a general architecture that can model additional classes. Another trend observed is the contrast between architectural paradigms: MARS, being autoregressive, consistently achieves higher recall, while segmentation-based models like those in SamRoad (Hetang et al., 2024) tend to yield lower recall but comparatively higher precision. Despite these differences, our model demonstrates superior balanced F1 scores across datasets, outperforming previous state-of-the-art approaches on average.

**Buildings.** To assess model performance on the building class, we utilize standard metrics such as Average Precision (AP), Average Recall (AR), and Intersection over Union (IoU), which are well-suited for evaluating polygonal predictions. We use AICrowd dataset V1 (AICrowd, 2020) that has been widely adopted in prior research. Table 4 summarizes our model's performance on AICrowd-V1. Similarly, MARS achieves closely on-par model performance with the previous state-of-the-arts models despite being a completely generic map generation model without any hypar-parameters.

Such performance highlights two major promising advantages of MARS framework: (1) **Simplicity**: MARS handles all diverse geometry shapes without any specific post-processing or tuned hyperparameter like previous works (Hetang et al., 2024; Wang et al., 2024), including vertex confidence threshold tuning, non-maximum-suppression IOU, etc. (2) **Scalability**: As MARS demonstrates it can learn various fundamental geometry features: polylines, multi-polylines, and polygons, the model posses great potential to scale from current two-class to multi-class model, and finer-grained classification, such as highway vs pedestrian way. This indicates that MARS can serve as the foundational model for an expanding range of future tasks related to map generation.

## 5.3 IMPORTANCE OF MAP-3M PRE-TRAINING

From our experiments, we find pretraining auto-regressive MARS will lead to significant faster convergence with a much higher accuracy e.g., from **70.45%** to **82.05%** on SpaceNet (SpaceNet, 2018) as shown in Table 5. This challenge appears specific to training auto-regressive models: Unlike traditional rasterized segmentation approaches, MARS relies solely on next-token prediction, making them more sensitive to data scarcity and harder to optimize from limited datasets. By contrast, with pre-training on MAP-3M, MARS quickly adapts and starts to pick-up target map features, highlighting the importance of large-scale pretraining for a foundational model.

Table 5: Effect of pretraining on downstream performance.

| Model | SpaceNet | | | AICrowd |
|---|---|---|---|---|
| | P | R | F1 | IOU |
| W/O-Pretrain | 77.62 | 64.48 | 70.45 | 95.09 |
| W-Pretrain | **79.68** | **84.56** | **82.05** | **95.24** |

## 5.4 CHAT WITH MARS

Model performance in real-world deployment can have out-of-domain generalization issues regardless of architectural advances. Existing systems inevitably face such domain gaps that hinder the end-to-end mapping performance. Chat-with-MARS, as outlined in Section 3, enables users to guide and refine the model's predictions, providing a more responsive and adaptive solution. To assess its effectiveness, we present formal evaluations based on two following protocols:

- **Chat with 1 point**: In this setting, a single GT vertex corresponding to the starting point of a missed road or building is introduced into a new SOS inference pass as defined in Sec. 3.1. This mimics the "one-click" user behavior, where a new minimal user input is used to guide the model in completing some missing map elements.

- **Chat with 2 points**: This configuration extends the protocol by providing two vertices of a missing object, forming a "two-click" setup that supplies a spatial direction. For both protocols, new predictions will be combined with old predictions for evaluation.

We quantitatively evaluate these two configurations for both road and building classes using the Cityscale (He et al., 2020a), SpaceNet (SpaceNet, 2018), and AICrowdV1-partial (AICrowd, 2020) datasets (the subset of VAL images with missing building elements). As reported in Table 6, incorporating user input consistently improves precision, recall, F1 score, and IoU across all datasets with diverse polygonal or polyline structures, relative to the baseline model. These results demonstrate the efficacy of interactive support in enhancing model accuracy and robustness, also highlighting promising opportunities for further exploration of human-in-the-loop map generation.

Table 6: Performance of Chat with MARS across different datasets.

| Model | CITYSCALE | | | SPACENET | | | AICROWD |
|---|---|---|---|---|---|---|---|
| | P | R | F1 | P | R | F1 | IOU |
| **MARS** | 84.28 | 81.53 | 82.88 | 79.68 | 84.56 | 82.31 | 97.32 |
| **Chat 1pt w MARS** | 84.95 | 82.66 | 83.79 | 81.40 | 84.65 | 82.95 | 97.35 |
| **Chat 2pts w MARS** | **85.15** | **83.14** | **84.13** | **81.74** | **85.12** | **83.17** | **97.40** |

## 5.5 LIMITATIONS

**Computational efficiency.** MARS's auto-regressive nature brings more computational overhead than the traditional segmentation-based one shot methods, mainly because autoregressive decoding is inherently sequential. However, there have been many auto-regressive model acceleration techniques that greatly speedup the model efficiency such as KV-cache acceleration, parallel decoding, etc. We hope our first work establishes the architecture prototype for more future works in map generation foundational research to improve both effectiveness and efficiency.

**Challenging case visualizations.** There are many challenging scenarios in our curated NAIP-3M dataset. These challenge cases can pose great difficulties for both segmentation-based post-processing vectorization and auto-regressive vectorization methods, therefore we'd like to highlight: (i) Complex Road Intersections: In certain cases, our MARS model can produce correct intersections; see Fig. 7. While in some cases, errors may occur when the model prioritizes the main roadway and overlooks thinner features. Representative failure cases are shown in Fig. 9 (a–c) in Appendix. A.3. (ii) Occluded Structures: heavy tree canopy or shadows can obscure building boundaries or road surfaces. Some failure cases are shown in Fig. 9 (d–f) in Appendix. A.3. While these cases can lead to missed vertices, we also include certain successful examples demonstrating that the model often learns to infer plausible shapes even under substantial occlusion. Examples of these cases are shown in Fig. 8 in the same section.

## 6 CONCLUSION

In this paper, we propose MARS: a foundational auto-regressive map generation framework with three major contributions: (1) we proposed map-to-sequence conversion algorithm to address map generation from a language-modeling perspective rather than visual perspective, a.k.a, treating maps as a foreign language; (2) we curated MAP-3M dataset that is $10\times$ larger than the current biggest map dataset to enable foundational model training; (3) finally, we present MARS that addresses map generation in an unified and end-to-end manner without any post-processing. The emerging "Chat with MARS" feature enables a brand new human-in-the-loop vectorized map generation capability.

**Disclaimer.** This research is supported by the National Geospatial-Intelligence Agency (NGA) via Contract No. HM04762491002. Any opinions, findings, conclusions or recommendations expressed in this material are those of the author(s) and do not necessarily reflect the views of NGA, DoD, or the US government. Approved for public release, NGA-U-2025-02299.

**Acknowledgements.** We would like to thank Raphael Tang, Xiaojie Guo, Mike Bianco, and Jacob Kovarskiy for their insightful feedback on the draft. We gratefully acknowledge support from Microsoft Spectre AI and Microsoft Planetary Computer.

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

## A  APPENDIX

### A.1  MAP-3M DATA CURATION

**Data Collection Process.** MAP-3M images are sourced from the National Agriculture Imagery Program (NAIP) (U.S. Department of Agriculture, 2025), known for its exceptional aerial imagery resolution at 0.6 meter per pixel (a small ratio of images can be of 0.3 or 1.0 meter per pixel), facilitating detailed feature extraction and accurate object delineation. Leveraging population data from United States Cities Database (United States Cities Database, 2025), we sample a total of 5,000 cities from 50 states, proportionally distributing based on state-level populations. The geographic distribution is shown in Figure 5 (a). From each city, we collect the most recent image chip (usually of size 10k by 10k pixels, or 20k by 20k pixels) from time range of 2020 - 2024. We further tile each big image chip into 512x512 subtiles without overlapping.

**Label Collection Process.** Building annotations in MAP-3M are sourced from the Overture Map Foundation (Overture, 2024), which currently has the most comprehensive building annotation across the globe by merging multiple authoritative and community-contributed datasets, including OpenStreetMap, Esri Community Maps, and machine-learning-derived building footprints from Google and Microsoft to fill in annotation gaps. Road annotations are sourced from OpenStreetMap (Map, 2017), encompassing all road subtypes such as highway, motorway, path, bridleway, etc., thereby ensuring comprehensive coverage and diverse structural representation. Spatial and statistical distributions of MAP-3M are shown in Figure 5, covering a wide-range of diverse scenes.

Table 7: Building performance on Aicrowd V2

| Model | AICROWD-V2 | | | | | | | | | | |
|---|---|---|---|---|---|---|---|---|---|---|---|
| | AP | AP50 | AP75 | AR | AR50 | AR75 | bAP | IoU | C-IoU | PoLiS | N-ratio |
| **MARS** | 30.91 | 57.06 | 30.07 | 70.03 | 86.31 | 73.85 | 49.84 | 74.42 | 66.49 | 4.135 | 1.012 |

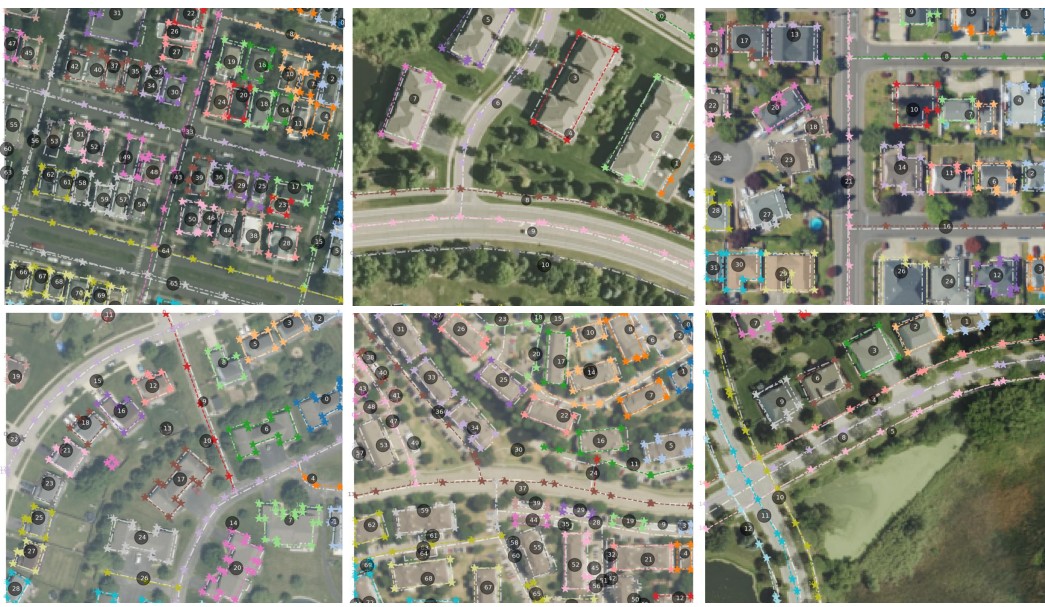

Figure 7: MARS visualization. Robust handling of complex road intersections by MARS. White dash-dotted line: Ground truth. Colored lines with asterisks: Predicted sequences. Black circles with white numbers: Object IDs. Source: MARS.

**Labeling Format.** Annotations within MAP-3M adopt a unified labeling format derived from the COCO standard (Lin et al., 2014), facilitating streamlined integration into existing machine learning frameworks. Buildings are treated as different objects, and are provided with dynamic-length vertex sequences consistent with the COCO format. In contrast, roads undergo a conversion process wherein each image's road network is initially represented as a flat graph, subsequently processed via our specialized stroke-based algorithms. The resultant separate road polylines are treated as different objects, with a difference that their vertex sequences are open-ended. This unified representation ensures compatibility and simplifies multi-class annotation handling in downstream tasks. Dataset has been released.

## A.2 AICROWDV2 VAL

To assess model performance on the building class, we utilize standard metrics such as Average Precision (AP), Average Recall (AR), and Intersection over Union (IoU), which are well-suited for evaluating polygonal predictions. Our evaluation spans two distinct test sets from the AICrowd dataset (AICrowd, 2020): version 1 (V1) and version 2 (V2). This dual evaluation approach is motivated by specific characteristics of the dataset. V1 has been widely adopted in prior research, but recent findings in Pix2Poly (Adimoolam et al., 2025) suggests potential data leakage between its training and validation splits. V2 addresses this issue with a revised structure, including new training and test splits. Notably, V1's validation set contains approximately 60,000 images, whereas V2's test set includes around 25,000 images. Due to lack of published benchmarks on V2, we believe our results represent the first public evaluation on this corrected version. Table 4 summarizes our model's performance on AICrowd-V1, while Table 7 presents results on AICrowd-V2.

## A.3 ADDITIONAL RESULTS

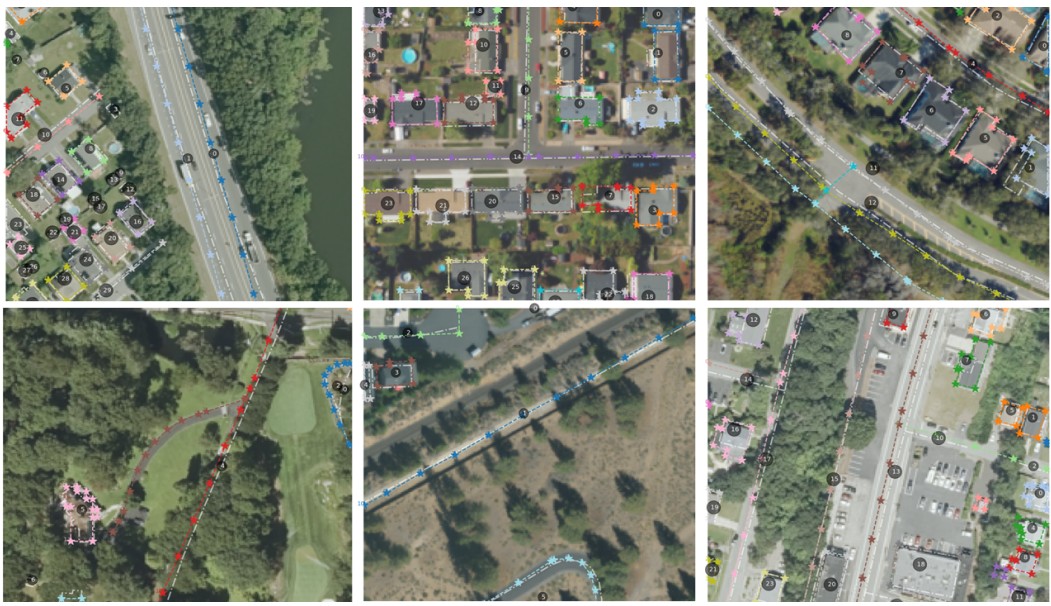

Figure 8: MARS visualization. Robust handling of occluded structures by MARS. White dash-dotted line: Ground truth. Colored lines with asterisks: Predicted sequences. Black circles with white numbers: Object IDs. Source: MARS.

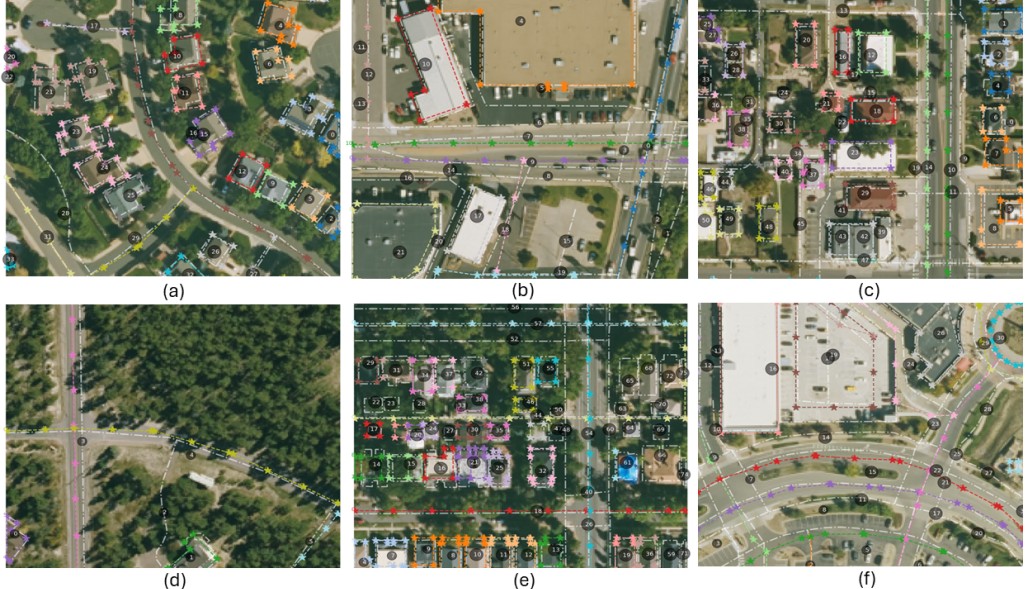

Figure 9: MARS visualization. Difficult examples like (a–c) complex intersections and (d–f) dense canopies occluding roads and buildings where MARS finds it difficult to preserve high accuracy. White dash-dotted line: Ground truth. Colored lines with asterisks: Predicted sequences. Black circles with white numbers: Object IDs. Source: MARS.

