# OpenReview forum: "MARS - A Foundational Map Auto-Regressor"
_ICLR.cc/2026/Conference — ICLR 2026 Poster_

### Official Review · Reviewer_3621 · 2025-10-27

**Soundness:** 3
**Presentation:** 3
**Contribution:** 3
**Rating:** 6
**Confidence:** 2

**Summary:**

This paper introduces MARS, the first foundational model for vectorized map generation that unifies the creation of both road networks and building polygons within a single end-to-end framework. MARS treats vectorized map primitives as a language and employs a sequence-to-sequence auto-regressive Transformer to directly generate map elements without intermediate steps. The model is trained on a newly curated large-scale dataset, MAP-3M, which contains three million high-quality, multi-class map annotations—10× larger and 100× broader in coverage than existing benchmarks. Extensive experiments demonstrate that MARS outperforms prior rasterization-based and hybrid approaches while maintaining scalability and generalization. Additionally, the authors propose “Chat with MARS,” an interactive human-in-the-loop system that enables prompt-based map generation and correction.

**Strengths:**

1. MARS  is the first foundational auto-regressive model for vectorized map generation, unifying both road networks and building polygons within a single end-to-end framework. The proposed map-to-sequence representation elegantly converts geometric primitives into a sequential language-like form, enabling map generation to benefit from advances in large-scale sequence modeling.
2. The work not only presents a new modeling paradigm but also releases MAP-3M, the largest multi-class map dataset to date, supporting robust training and reproducibility. Extensive experiments demonstrate strong generalization and consistent performance gains over prior rasterization-based and hybrid approaches.
3. The “Chat with MARS” module creatively leverages prompt-following capabilities of the auto-regressive model, introducing a novel human-in-the-loop mechanism for real-time map editing and correction. This interactivity significantly enhances the paper’s applicability to real-world geospatial workflows.

**Weaknesses:**

While the proposed map-to-sequence formulation is elegant, it inevitably flattens geometric structures into linear token sequences. The model relies mainly on data-driven regularities rather than structural constraints.

**Questions:**

While MARS directly generates vectorized map elements through an end-to-end sequence model, it remains unclear how the proposed approach ensures geometric or topological boundary consistency between adjacent objects. Does the model incorporate any explicit mechanism or loss to preserve boundary alignment, or is this consistency purely learned implicitly from data?

---

> ### Author Response · Authors · 2025-11-21
> **Reply to Reviewer #4**
>
> We appreciate the reviewer’s positive review and thoughtful questions. Our responses are provided below.
>
>
> **Q1: While MARS directly generates vectorized map elements through an end-to-end sequence model, it remains unclear how the proposed approach ensures geometric or topological boundary consistency between adjacent objects. Does the model incorporate any explicit mechanism or loss to preserve boundary alignment, or is this consistency purely learned implicitly from data?**
>
>
> **A1:** We appreciate the reviewer’s insightful comment.
>
> - Consistency is learned implicitly through both data-driven regularities and structural constraints encoded in the image features. Boundary consistency emerges naturally from the autoregressive formulation: each next token learning is conditioned on all previously generated tokens, which are fully contextualized through cross-attention with all image tokens in the transformer decoder. The structural constraints present in the image embeddings (e.g., building edges, road boundaries, shared contours) propagate through the decoder and help the model learn coherent geometric and topological relationships between adjacent objects.
>
>
> - Visual evidence included.
> We provide qualitative examples on the project website (please see the link:https://huggingface.co/spaces/bag-lab/MARS) demonstrating that the model often produces well-aligned boundaries across nearby objects in practice.
>
>
> **Q2: While the proposed map-to-sequence formulation is elegant, it inevitably flattens geometric structures into linear token sequences. The model relies mainly on data-driven regularities rather than structural constraints.**
>
> **A2:** We thank the reviewer for the thoughtful comment. The consistency is learned implicitly from both data-driven regularities and the structural constraints encoded in the image features. Each next token learning is conditioned on all previously generated tokens, and tokens attend to all image tokens through cross-attention. The structural constraints present in the image features (such as road boundaries, building edges, and shared contours) are injected into every decoding step via cross-attention, guiding the model to produce geometrically and topologically aligned objects in practice.

---

> > ### Comment · Reviewer_3621 · 2025-11-22
> >
> > Thank you for the detailed clarification and examples. Much appreciated!

---

### Official Review · Reviewer_nRw3 · 2025-10-28

**Soundness:** 3
**Presentation:** 3
**Contribution:** 3
**Rating:** 4
**Confidence:** 3

**Summary:**

This paper introduces a novel foundation model for map generation, termed MARS. The proposed approach employs a ViT backbone to extract visual features and an autoregressive transformer to generate map sequences, including points, polylines, and polygons. In addition, the authors present a multi-class map dataset to support the foundation model and benchmarking. The paper also introduces an interactive feature, Chat-with-MARS, which allows human-in-the-loop map generation and correction.

**Strengths:**

1. The paper presents MARS, a novel foundation model for vectorized map generation, which is both original and technically sound.
2. The MARS framework demonstrates strong performance.
3. The proposed Chat-with-MARS functionality is innovative and adds practical value by enabling interactive map generation.

**Weaknesses:**

1. While the authors provide detailed definitions of different map objects, autoregressive generation typically relies on a well-defined sequence order. However, the ordering of map objects in the proposed pipeline is not clearly explained.
2. The use of a ViT-based visual backbone and an autoregressive transformer could make the model computationally expensive. The paper lacks discussion or analysis regarding computational efficiency or runtime performance.
3. As a foundational model for map generation, the ablation studies are insufficient. The authors do not adequately examine the effects of architectural choices (e.g., different vision backbones or decoder designs) or training strategies.
4. Although the Chat-with-MARS feature is compelling, the paper would benefit from additional demonstrations—such as a short demo video or a GUI prototype—to better illustrate its capabilities and user interaction.

**Questions:**

Please refer to the weaknesses.

---

> ### Author Response · Authors · 2025-11-21
> **Reply to Reviewer #3**
>
> We thank the reviewer for the thoughtful feedback and constructive suggestions. Our responses are provided below.
>
>
> **Q1: While the authors provide detailed definitions of different map objects, autoregressive generation typically relies on a well-defined sequence order. However, the ordering of map objects in the proposed pipeline is not clearly explained.**
>
> **A1:**  We appreciate the reviewer’s comment and would like to clarify:
> - We agree that sequence order is important for autoregressive learning. In our pipeline, we sort all map objects by their distance to the image centroid, and when distances are similar, we sort them by their clockwise angle.
>
> - This creates a consistent spiral ordering across the dataset and ensures reproducible decoding.
>
> - We will clarify this procedure in the revised manuscript, adding the description to Section 2.2- Map Auto-Regressor Architecture (last paragraph).
>
>
> **Q2: The use of a ViT-based visual backbone and an autoregressive transformer could make the model computationally expensive. The paper lacks discussion or analysis regarding computational efficiency or runtime performance.**
>
> **A2:** We appreciate the reviewer’s observation and would like to clarify:
> - The current design introduces notable computational cost, largely because autoregressive decoding is inherently sequential; AR models are indeed slower than one-shot segmentation methods by nature.
>
> - Our work lays the architecture foundations for future work in KV-cache acceleration, parallel decoding, and lighter or distilled visual encoders.
>
> - We will include this discussion in the Limitations subsection of the Experimental Results section in the revised manuscript.
>
> **Q3: As a foundational model for map generation, the ablation studies are insufficient. The authors do not adequately examine the effects of architectural choices (e.g., different vision backbones or decoder designs) or training strategies.**
>
> **A3:** Thank you for the comment. We have added ablation studies on both encoder variants (Swin-L vs. SAM-B) and decoder depth, evaluated on the same validation set. The results show two key trends:
> Decoder depth: Performance improves as we increase the number of transformer blocks, with gains plateauing around a median depth of 8.
> Backbone choice: Larger backbones such as Swin-L consistently outperform SAM-B on both road and building detection for the same decoder depth. Swin-L provides stronger multi-scale features, leading to better IoU for buildings and slightly higher F1 for roads compared to SAM-B.
>
> | Vision Backbone | Decoder Depth | P (road) | R (road) | F1 (road) | IoU (building) |
> |-----------------|----------------|----------|----------|-----------|------------------|
> | Swin-L          | 4              | 89.9     | 74.5     | 81.5      | 57.8             |
> | Swin-L          | 8              | 89.5     | 78.0     | 83.4      | 61.1             |
> | Swin-L          | 12             | 89.7    | 77.9     | 83.4      | 61.8             |
> | SAM-B           | 4              | 90.3    | 73.9     | 81.3      | 53.6            |
> | SAM-B           | 8              | 90.4        | 75.6        | 82.4         | 55.3                |
> | SAM-B           | 12             | 90.1        | 76.6        | 82.8         | 55.8                |
>
>
> **Q4: Although the Chat-with-MARS feature is compelling, the paper would benefit from additional demonstrations—such as a short demo video or a GUI prototype—to better illustrate its capabilities and user interaction.**
>
> **A4:** Thank you for this helpful suggestion. We have now set up a project website that includes short demo videos of Chat-with-MARS in different interaction modes, MARS generative-style inference. Please see the “Click demo” section on the project webpage: https://huggingface.co/spaces/bag-lab/MARS

---

### Official Review · Reviewer_SQaC · 2025-10-30

**Soundness:** 3
**Presentation:** 3
**Contribution:** 2
**Rating:** 6
**Confidence:** 2

**Summary:**

This paper proposes MARS, an auto-regressive foundational model for end-to-end vectorized map generation that unifies the prediction of roads (as multi-polylines) and buildings (as polygons) without relying on post-processing heuristics. The authors introduce a novel map-to-sequence representation, a large-scale multi-class dataset (MAP-3M), and a human-in-the-loop interaction paradigm called “Chat with MARS.” Sounds good.

**Strengths:**

1. This research topic is new to me.
2. The writing is good.
3. Interactive capability is intersting.

**Weaknesses:**

1. No ablation on the impact of the stroke-based decomposition vs. alternative graph traversal or serialization strategies
2. Limited analysis of failure modes (e.g., complex intersections, occluded structures).
3. The “Chat with MARS” evaluation is synthetic (uses GT points as prompts); real-user studies or robustness to noisy clicks would strengthen claims.

**Questions:**

1. Can the dataset be generalized to global regions with different road/building styles?

---

> ### Author Response · Authors · 2025-11-21
> **Reply to Reviewer #2**
>
> We thank the reviewer for the positive review and insightful questions. Our responses are provided below.
>
> **Q1: No ablation on the impact of the stroke-based decomposition vs. alternative graph traversal or serialization strategies.**
>
>  **A1:** We appreciate the reviewer’s comment and would like to clarify:
>
> - Stroke decomposition is information-lossless and preserves true topology.
> Stroke decomposition is a well-established cartographic standard (Thomson & Richardson, 1999) designed to follow the principle of “good continuation” and align with how humans perceive road networks. At T-junctions, where roads meet but do not cross, no new node is added. At intersections, all existing junction nodes are preserved without removal or modification.
>
> - In contrast, traversal-based strategies such as DFS or BFS ignore geometric continuity, often generating order-dependent, semantically meaningless sequences that break real-world road structure.
>
> - We inspected many examples of difficult cases, including intersections, roundabouts, T junctions, and split roads, and found that the stroke algorithm consistently produces correct, topology preserving decompositions. Please see the “Stroke Visualizations” section on the project webpage:https://huggingface.co/spaces/bag-lab/MARS
>
> **Reference:** Thomson, Robert C., and Dianne E. Richardson. "The ‘good continuation’principle of perceptual organization applied to the generalization of road networks." Proceedings of the ICA 19th international cartographic conference. Ottawa, Canada, 1999.
>
> **Q2: Limited analysis of failure modes (e.g., complex intersections, occluded structures).**
>
> **A2:** We provide a set of visual examples on our project website (Please see the “Challenging Cases” section on the project webpage:https://huggingface.co/spaces/bag-lab/MARS). The main categories of failures we observe are:
>
> - Complex intersections: Errors occur when the model prioritizes the main roadway and overlooks thinner features, the visuals also include more correct intersection examples.
>
> - Occluded structures: Heavy tree canopy or shadows can obscure building boundaries or road surfaces. While these cases can lead to missed vertices, we also include more successful examples demonstrating that the model often learns to infer plausible shapes even under substantial occlusion.
>
> We will include this analysis in the revised manuscript, adding the discussion to the Limitations subsection of Experimental Results section and the corresponding visual examples to Appendix A.3.
>
> **Q3: The “Chat with MARS” evaluation is synthetic (uses GT points as prompts); real-user studies or robustness to noisy clicks would strengthen claims.**
>
> **A3:** We appreciate this suggestion and have analyzed Chat-with-MARS's robustness to click position noise. Visual demonstrations showing results across multiple noisy click inputs are provided in the 'Click Jitter Analysis' section on our project website: https://huggingface.co/spaces/bag-lab/MARS"
>
> **Q4: Can the dataset be generalized to global regions with different road/building styles?**
>
> **A4:** MAP-3M was designed as a large-scale U.S.-focused dataset, covering diverse geographic and urban contexts across 49 states. While its primary scope is the U.S., the following points address generalization:
>
> - Transferability: Models trained on MAP-3M can serve as a pretraining resource for global applications. Fine-tuning on regional datasets can adapt the learned representations to local styles, which is a common practice in geospatial modeling. It is beneficial for downstream tasks.
>
> - Diversity Within U.S. Regions: The dataset spans rural, suburban, and dense urban areas, capturing a wide range of road networks and building architectures. This diversity provides a strong foundation for models to learn robust features.
>
> - Future Extensions: We recognize the importance of global coverage and plan to extend MAP-3M with additional regions in future releases. The current release focuses on quality and scale within the U.S. to establish a strong baseline.

---

### Official Review · Reviewer_rikJ · 2025-10-30

**Soundness:** 3
**Presentation:** 3
**Contribution:** 3
**Rating:** 6
**Confidence:** 4

**Summary:**

This paper proposes MARS, an end-to-end map auto-regressor for vectorized map generation. The key idea is to treat maps as a foreign language: all vector primitives (points, polylines/roads, polygons/buildings) are serialized via a map-to-sequence procedure, and then a vision encoder + autoregressive transformer decodes the whole map token by token. On top of that, the authors introduce “Chat with MARS”, an interactive, human-in-the-loop decoding mode (start-/mid-/end-of-sequence interventions) that can fix missing or drifting map objects with 1–2 clicks. To support this, the paper also curates MAP-3M, which is the largest aerial-image + multi-class (roads + buildings) dataset so far (∼3M tiles, wide US coverage, NAIP imagery + Overture/OSM labels). Experiments on Cityscale, SpaceNet, and AICrowd show that the unified, class-agnostic, autoregressive model is competitive with or close to SOTA methods that are narrowly specialized.

**Strengths:**

* Framing vectorized map generation as AR sequence modeling is elegant: one decoder, one vocabulary, one loss, multiple geometry types. This is nicer than the usual “segmentation → heuristic post-processing” pipeline.
* Even though the current demos look a bit toy-ish, the idea of AR decoding + teacher forcing ⇒ promptability is solid, and the paper shows three concrete intervention modes (SOS/MOS/EOS) with quantitative gains. This is, in my view, the most future-facing part.
* A 3M-tile, dual-class, US-wide, reasonably high-res dataset that already comes vectorized would be very valuable to the community, especially for people doing OSM updating, change detection, and AD map pretraining. This alone can justify publication if it’s really as large, clean, and diverse as stated.
* On Cityscale / SpaceNet they get very reasonable TOPO F1s, sometimes higher recall than road-specialized models, which is non-trivial for a generic AR model.

**Weaknesses:**

* Right now the two main claims — “we have a 3M, dual-class, high-quality dataset” and “we have a working general AR map model” — cannot be verified. In the last year I have seen many papers claim “large-scale dataset” and the final release was (i) much smaller, (ii) missing one of the modalities, or (iii) under a restrictive license. So the impact is contingent on actual release.
* The whole “maps as a sequence” story stands or falls with the correctness of the stroke decomposition at intersections, roundabouts, and T-junctions. If this step introduces topology errors or weird ordering, the model will happily learn those artifacts. This should be stress-tested more.
* The 1-click / 2-click “Chat with MARS” is impressive, but in the current form it is still a single-object recovery tool. For real OSM editing or AD map maintenance, users will want multi-round constraints (e.g., “keep these 3 roads, regenerate everything north of x”), not only point hints.
* I suggest citing some downstream application papers like P-MapNet: Far-seeing map generator enhanced by both SDMap and HDMap priors.

**Questions:**

* Will MAP-3M be truly public, with both imagery and vector labels, or only labels? Are there licensing constraints that will make the “3M” number lower in practice?

---

> ### Author Response · Authors · 2025-11-21
> **Reply to Reviewer #1**
>
> We thank the reviewer for the positive and detailed review as well as the suggestions for improvement. Our responses are provided below:
>
> **Q1: Will MAP-3M be truly public, with both imagery and vector labels, or only labels? Are there licensing constraints that will make the “3M” number lower in practice?**
>
>  **A1:** We appreciate the reviewer’s concern and would like to clarify:
>
> - Public Release & Licensing: The MAP-3M dataset has already been released as an open public dataset on Hugging Face under a Creative Commons license for research purposes. It is freely accessible at: https://huggingface.co/datasets/bag-lab/MAP-3M.
>
> - Dataset Scale & Coverage: MAP-3M contains approximately 3 million high-resolution images, covering a large geographic area of ~294,069 km² across all 49 U.S. states. This ensures the scale and diversity claimed in the paper.
>
> - Modalities & Annotations: The dataset includes both imagery and vector labels. Annotations follow COCO-style format, and currently support two classes: roads and buildings. These modalities are fully available in the released version.
>
> - No Licensing Constraints: There are no restrictions that would reduce the dataset size or limit access. The full dataset, as described in the paper, is publicly available for research efforts.
>
> **Q2; The whole “maps as a sequence” story stands or falls with the correctness of the stroke decomposition at intersections, roundabouts, and T-junctions. If this step introduces topology errors or weird ordering, the model will happily learn those artifacts. This should be stress-tested more.**
>
> **A2:** Thank you for the suggestion, and we would like to clarify the following
>
> - **Stroke decomposition is information-lossless and preserves true topology**
>
> Stroke decomposition is a well-established cartographic standard (Thomson & Richardson, 1999) designed to follow the principle of “good continuation” and align with how humans perceive road networks. At T-junctions, where roads meet but do not cross, no new node is added. At intersections, all existing junction nodes are preserved without removal or modification.
>
> - **Only the ordering of objects changes, not their structure**
>
> Stroke decomposition may change the per-object sequence order, but this is orthogonal to topology. In our pipeline, map objects are ultimately re-sorted in a deterministic global order (e.g., spiral ordering) purely to standardize input format and stabilize training—without modifying connectivity or geometry.
>
> - **Extensive stress-testing shows no topology loss**
>
> We inspected many examples of difficult cases (multi-leg intersections, roundabouts, T-junctions, dual-carriageway splits, etc.). In all scenarios, the decomposition consistently preserves connectivity, does not drop intersection nodes, and does not introduce unexpected artifacts. Please see the “Stroke Visualizations” section on the project webpage:https://huggingface.co/spaces/bag-lab/MARS
>
> **Reference:** Thomson, Robert C., and Dianne E. Richardson. "The ‘good continuation’principle of perceptual organization applied to the generalization of road networks." Proceedings of the ICA 19th international cartographic conference. Ottawa, Canada, 1999.
>
> **Q3; The 1-click / 2-click “Chat with MARS” is impressive, but in the current form it is still a single-object recovery tool. For real OSM editing or AD map maintenance, users will want multi-round constraints (e.g., “keep these 3 roads, regenerate everything north of x”), not only point hints.**
>
> **A3:** Thank you for recognizing the Chat-with-MARS features and for the constructive feedback. We have now set up a project website that includes short demo videos of Chat-with-MARS in different interaction modes. Please see the “Click demo” section on the project webpage: https://huggingface.co/spaces/bag-lab/MARS. We are expanding support for multi-round constraints and will update the visuals accordingly.
>
> **Q4: I suggest citing some downstream application papers like P-MapNet: Far-seeing map generator enhanced by both SDMap and HDMap priors.**
>
> **A4:** Thank you for the helpful suggestion. P-MapNet: Far-seeing Map Generator Enhanced by Both SDMap and HDMap Priors is an important contribution to the field, and we will cite and contextualize it appropriately in the revised manuscript.

---

> > ### Comment · Reviewer_rikJ · 2025-11-27
> > **feedback**
> >
> > Thanks for the detailed and clarifying response. I appreciate the confirmation that the MAP-3M dataset is indeed publicly released and that both imagery and vector annotations are available under a permissive license, that is excellent for reproducibility and future research. The additional explanation about stroke decomposition and topology preservation is also convincing and well-supported by references.
> >
> > Overall, I think this is a novel and meaningful contribution that not only builds valuable dataset infrastructure but also explores an important emerging direction: applying large-model paradigms to structured geometric data. The work has clear potential for real applications such as high-definition map maintenance in autonomous driving.
> >
> > However, it seems the authors have not yet uploaded a revised PDF reflecting the cited updates. I recommend submitting an updated version so that these improvements are properly reflected in the camera-ready or revision stage.

---

### Author Response · Authors · 2025-12-03
**Summary response**

Thanks to all reviewers for the valuable feedback.

We have updated our manuscript to include the following discussions:

- For reviewer rikJ’s suggestion regarding citing downstream application papers, we have cited the paper  P-MapNet: Far-seeing map generator enhanced by both SDMap and HDMap priors in the paragraph in Sec1 Introduction: "
A key challenge for map generative modeling lies in the vectorized representation of maps (Congalton,1997;Jiangetal.,2024)."

- For reviewer SQaC's suggestion to include an analysis of failure modes, we have added the paragraph in section 5.5 Limitations: "Challenging Case Visualizations". We have added corresponding visual examples to Appendix A.3: "ADDITIONAL RESULTS".

- For reviewer nRw3's question regarding the unclear explanation of the ordering of map objects in the proposed pipeline, we have added the clarification in Sec 2.2 MAP AUTO-REGRESSOR ARCHITECTURE: "Additionally, we sort all map objects by their distance to the image centroid, and when distances are similar, we sort them by their clockwise angle. This creates a consistent spiral ordering across the dataset and ensures reproducible decoding.".

- For reviewer nRw3's suggestion to add a discussion of computational efficiency, we have added the paragraph in section 5.5 Limitations: "Computational Efficiency".

- We have added the demo page (https://huggingface.co/spaces/bag-lab/MARS) and include more interactive results to address reviewers rikJ, SQaC, nRw3's concerns on correctness and robustness of stroke decomposition, the capabilities and robustness of "Chat with MARS” and the analysis of failure modes.

- We have added the MAP-3M dataset link (https://huggingface.co/datasets/bag-lab/MAP-3M) to address reviewers rikJ's concerns about the dataset’s licensing and completeness.

We hope this addresses reviewers' concerns. Thanks again.

---

### Meta-Review · Area_Chair_L3te · 2026-01-06

**Summary:**

This paper proposes MARS, an end-to-end map autoregressor for vectorized map generation, as well as the MAP-3M dataset, which is currently the largest aerial-image-based, multi-class dataset of its kind. The initial score was 6/6/6/4, and the score remained unchanged before the reset.

The reviewers’ initial concerns mainly focused on whether the dataset would eventually be made public, stress testing on complex terrains, analysis of failure modes, evaluation of the model’s computational cost, and the need for more extensive visualizations. The rebuttal successfully addressed most of these concerns and received positive feedback from multiple reviewers (rikJ, 3621) before the reset.

For the remaining two reviewers’ comments, the authors also provided relatively thorough responses, along with additional ablation studies and more visualizations. One reviewer (nRw3), who initially gave a score of 4, raised concerns regarding efficiency. Although this issue was not directly resolved in the rebuttal, efficiency is not the dominant factor in the target scenarios, and it can be mitigated by existing complementary methods.

After a careful assessment of the submission, reviews, response, and the discussion, the AC recommends acceptance. The authors are encouraged to further refine the paper by incorporating the reviewers’ valuable suggestions.

**Reviewer Concerns:**

The reviewers’ initial concerns mainly focused on whether the dataset would eventually be made public, stress testing on complex terrains, analysis of failure modes, evaluation of the model’s computational cost, and the need for more extensive visualizations. The rebuttal successfully addressed most of these concerns and received positive feedback from multiple reviewers (rikJ, 3621) before the reset.

For the remaining two reviewers’ comments, the authors also provided relatively thorough responses, along with additional ablation studies and more visualizations. One reviewer (nRw3), who initially gave a score of 4, raised concerns regarding efficiency. Although this issue was not directly resolved in the rebuttal, efficiency is not the dominant factor in the target scenarios, and it can be mitigated by existing complementary methods.

**Reviewer Scores:**

The manuscript received initial review scores of 6/6/6/4. After the rebuttal/discussion and before the reset, the score remained unchanged.

Since most concerns from the Reviewers are addressed after the rebuttal (see 'Reviewer Concerns'), I would approximate 8 (Reviewer rikJ )/6/6 (Reviewer nRw3)/6 as the final score.

---

### Decision · Program_Chairs · 2026-01-26

Accept (Poster)